# Implementation of a Modified Faster R-CNN for Target Detection Technology of Coastal Defense Radar

**He Yan \*, Chao Chen, Guodong Jin, Jindong Zhang, Xudong Wang and Daiyin Zhu**

College of Electronic and Information Engineering, Nanjing University of Aeronautics and Astronautics, Nanjing 210016, China; chenchao@nuaa.edu.cn (C.C.); jingguodong@nuaa.edu.cn (G.J.); jdzhang@nuaa.edu.cn (J.Z.); xudongwang@nuaa.edu.cn (X.W.); zhudy@nuaa.edu.cn (D.Z.)
\* Correspondence: yanhe@nuaa.edu.cn

**Abstract:** The traditional method of constant false-alarm rate detection is based on the assumption of an echo statistical model. The target recognition accuracy rate and the high false-alarm rate under the background of sea clutter and other interferences are very low. Therefore, computer vision technology is widely discussed to improve the detection performance. However, the majority of studies have focused on the synthetic aperture radar because of its high resolution. For the defense radar, the detection performance is not satisfactory because of its low resolution. To this end, we herein propose a novel target detection method for the coastal defense radar based on faster region-based convolutional neural network (Faster R-CNN). The main processing steps are as follows: (1) the Faster R-CNN is selected as the sea-surface target detector because of its high target detection accuracy; (2) a modified Faster R-CNN based on the characteristics of sparsity and small target size in the data set is employed; and (3) soft non-maximum suppression is exploited to eliminate the possible overlapped detection boxes. Furthermore, detailed comparative experiments based on a real data set of coastal defense radar are performed. The mean average precision of the proposed method is improved by 10.86% compared with that of the original Faster R-CNN.

**Keywords:** target detection; deep learning; constant false-alarm rate (CFAR); Faster R-CNN; coastal defense radar

## 1. Introduction

The ocean occupies approximately 71% of the total surface area of the Earth and has rich biological and mineral resources [1–3]. To prevent the invasion of territorial sea and illegal marine operations, the management of the domestic sea area must be strengthened. The important part of marine management is the monitoring of ships on the sea surface. Traditional radar target detection methods deal mainly with time, frequency, and transform domains [4,5], and improve the signal-to-clutter ratio to achieve reliable target detection [6]. In recent years, the rapid development of deep learning technology has enabled domestic and foreign scholars to put forward new target detection methods.

The AlexNet [7] convolutional neural network (CNN) architecture was the champion of the ImageNet challenge competition in 2012. Since then, many excellent neural network algorithms have emerged, such as the VGG [8], Inception [9], and ResNet [10], which constantly set records and set off waves of artificial intelligence. In the work of Girshick et al. [11], the region-based convolutional neural network (R-CNN) algorithm was proposed to use a CNN for target detection for the first time. It extracts image features using a deep convolutional network and uses the method of "selective search + CNN + SVM" for detection. The training process of the R-CNN needs much time and hard disk space. In response, Girshick proposed the Fast R-CNN algorithm [12] to overcome this problem. This algorithm uses feature sharing in region proposal and adds a region of interest pooling layer in the last convolution layer. However, both the R-CNN and Fast R-CNN are not complete end-to-end target detection systems because they rely on the proposed methods of external

regions outside the training stage. Consequently, the Faster R-CNN came into being. The Faster R-CNN [13] realizes the sharing of an image feature map between the region proposal network (RPN) and the target detection network. At the same time, the framework integrates region proposal generation, depth feature extraction, target recognition, and detection into a neural network model, which greatly improves the training efficiency and detection accuracy.

For the problem of the sea-surface target detection, to correctly distinguish the target and the reef, effectively suppress the clutter, and improve the detection accuracy, the typical differences among target, reef, and clutter must be found. This study investigates the application of the Faster R-CNN to detect two-dimensional sea-surface targets in the coastal defense radar [14] image. The Faster R-CNN extracts different input image features through multiple convolutions and inputs the feature information into the RPN to detect the target and locate the feature area. The filtered region proposals and the feature map output from the convolution layer are both used for end-to-end training to accurately obtain the target category and location information. In this study, the detection results based on the Faster R-CNN are compared with those of the traditional constant false-alarm rate (CFAR) target detection methods, such as the cell-averaging constant false-alarm rate (CA-CFAR), greatest-of cell-averaging constant false-alarm rate (GOCA-CFAR), and smallest-of cell-averaging constant false-alarm rate (SOCA-CFAR) [15–17]. Then, the advantages and disadvantages of these algorithms are analyzed. Aiming at the characteristics that the data set contains small targets and the resolution of the coastal defense radar is lower than that of the SAR image [18,19], we make some improvements based on the original Faster R-CNN, which help to improve the detection accuracy. The main contributions are listed as follows in more detail:

(1) The deep learning method is applied to the measured data of the coastal defense radar with a low resolution, and its detection results are compared with the two–dimensional CFAR detectors.
(2) ResNet50 is replaced with a relatively shallow backbone feature extraction network VGG16 in the Faster R-CNN, and a parametric rectified linear unit (PReLU) function is used to achieve more specialized activations.
(3) The K-means clustering algorithm [20,21] is added to calculate the size of the anchors in the RPN to better meet the data set characteristics and speed up the network convergence.
(4) The Soft-NMS [22] algorithm is used to eliminate the possible overlapped detection boxes.

## 2. Methods

As a traditional target detection method, CFAR has been widely used in real world applications. Therefore, this section mainly introduces the representatives of the CFAR detector, such as CA-CFAR, GOCA-CFAR, and SOCA-CFAR. Then, a new target detection method based on the Faster R-CNN is introduced. On this basis, we discuss the related work inspiring our method.

### 2.1. CFAR Detector

The CFAR detection of radar targets in clutter plus noise background is an important unit in radar signal processing. In this section, we discuss three different CFAR detectors and verify their performance in the following experiments.

### 2.1.1. CA-CFAR

The CA-CFAR detector is one of the most representative CFAR detection algorithms. The corresponding detection threshold is obtained by averaging the interference power of the adjacent cells around the cell under test (CUT); hence, it is named CA-CFAR detection. The algorithm structure is simple. The detection performance is good when the background is a uniform Rayleigh clutter. However, when multiple false targets exist in the reference

cells, the power of the false targets will raise the decision threshold of the CA-CFAR detector and reduce the detection probability, resulting in a missed detection of the real targets. The detection sliding window includes the CUT, protection cells, and reference cells.

Figure 1 shows the implementation structure of the detector, where $D$ is the echo power of the CUT, and $x_L = \frac{1}{n}\sum_{i=1}^{n} x_i$ and $x_R = \frac{1}{n}\sum_{i=1}^{n} x_{n+i}$ are the average interference powers of the reference cells on both sides of the CUT. The total number of reference cells is $N = 2n$. The protection cells prevent the target in the reference cells from raising the detection threshold. $\alpha$ is the nominal factor; $\beta^2$ is an estimate of the interference power in the sliding reference window cells, which can be obtained using the sample average of known data. The calculation formula is

$$\beta^2 = \frac{1}{2}(X_L + X_R) = \frac{1}{N}\sum_{i=1}^{N} X_i. \tag{1}$$

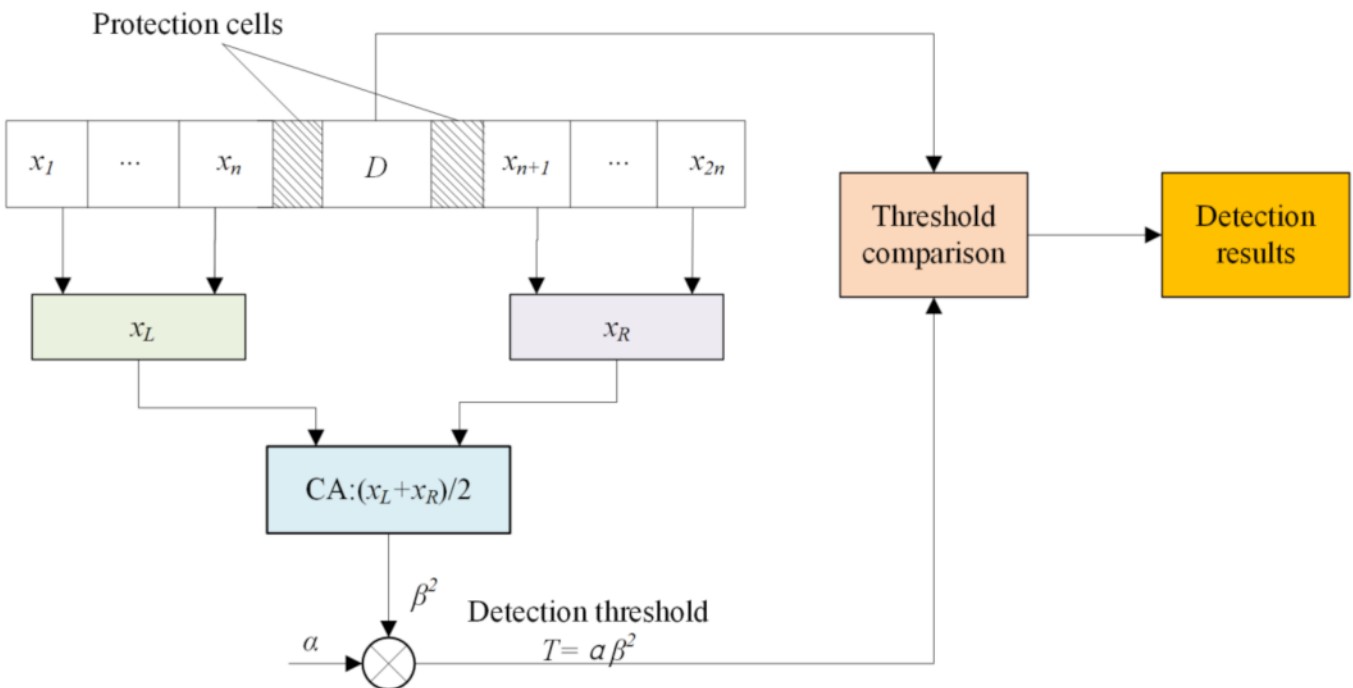

**Figure 1.** Cell average constant false-alarm rate (CA-CFAR) detector.

Given the false-alarm rate $P_{fa}$ and the number of reference cells $N$, the nominal factor $\alpha$ can be calculated as

$$\alpha = N\left(P_{fa}^{-1/N} - 1\right). \tag{2}$$

Therefore, the final detection threshold is obtained as follows:

$$T = \left(P_{fa}^{-1/N} - 1\right)\sum_{i=1}^{N} X_i. \tag{3}$$

In this paper, the two-dimensional CFAR method is used. Figure 2 shows the design method of the two-dimensional sliding reference window. The CUT is located in the center of the rectangular reference window, as shown in the red box in Figure 2. Noise power is computed from cells that contain no target signal. These cells are known as the reference cells. Reference cells form a band around the CUT but may be separated from the CUT by a protection band. In Figure 2, the number of rows and columns of the protection band cells on each side of the CUT is set as $N_{pr}$ and $N_{pc}$, respectively, and the protection band cells are indicated by the orange boxes. Similarly, the number of rows and columns of the

reference band cells on each side is set as $N_{rr}$ and $N_{rc}$, respectively, and the reference band cells are indicated by the green boxes. The total number of cells in the combined reference region, protection region, and CUT is $N_{total} = (2N_{rc} + 2N_{pc} + 1)(2N_{rr} + 2N_{pr} + 1)$. The number of reference cells is $N_{ref} = N_{total} - (2N_{pr} + 1)(2N_{pc} + 1)$.

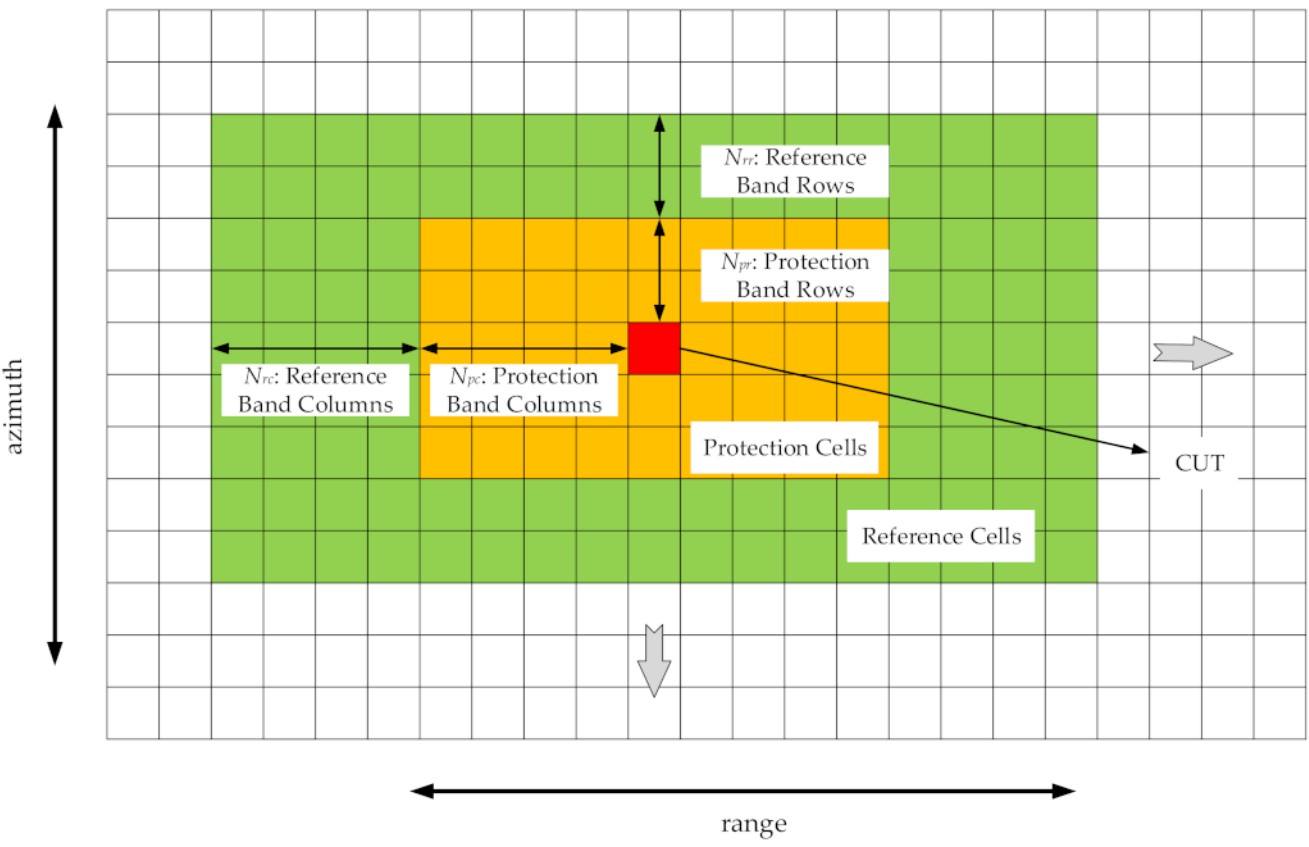

**Figure 2.** The structure of the two-dimensional sliding reference window.

The principle of the CA-CFAR detector shows that it has a good detection performance in a homogeneous clutter background but is not ideal under a multi-target and clutter edge environment. Moreover, when false targets exist in the reference cells, their power will raise the decision threshold of the CA-CFAR detector, resulting in an interference to the real target decision.

### 2.1.2. GOCA-CFAR and SOCA-CFAR

The GOCA-CFAR and SOCA-CFAR are improved methods of CA-CFAR. They are different from CA-CFAR in that they split the two-dimensional reference window around the CUT into left and right halves. Then, they calculate the sample mean for each half and select the greatest or smallest mean. The detailed description of GOCA-CFAR and SOCA-CFAR can be found in [16,17].

### 2.2. Target Detection Based on FASTER R-CNN

The traditional ship target detection algorithm in the complex scene radar image faces the problem of a high false-alarm rate and a low recognition accuracy rate. We introduce herein a target detection algorithm based on the Faster R-CNN. By training the measured data provided by the coastal defense radar, we can obtain a new target detector that can detect the accurate position of ships in different sea areas and interference backgrounds. This detector has excellent anti-clutter and anti-jamming ability.

### 2.2.1. ResNet50 Feature Extraction Network

ResNet stands for residual network. It is part of the backbone feature extraction network used widely in target classification and computer vision tasks. The main function of the traditional CNN is to extract the feature information of the input image. The CNN gradually extracts the underlying features to the highly abstract features. Therefore, the deeper the network, the more abstract the features extracted and the richer the semantic information. However, simply increasing the number of network layers will lead to problems of gradient disappearance and explosion and network degradation, making the overall network performance worse than that with fewer layers. ResNet creatively adds identity mapping to ensure that the performance of the deep network is equal to that of the shallow network when deepening the network. In the residual block structure shown in Figure 3, the right connecting line jumps before the activation function, and the output of the upper layer is added with the output of this layer, such that the summation result is input into the activation function as the final output of this layer. The residual structure can make the input information directly transmit to the output and protect the integrity of the information. Moreover, the network only needs to learn the difference between input and output, simplifying the learning objectives and reducing the learning difficulty.

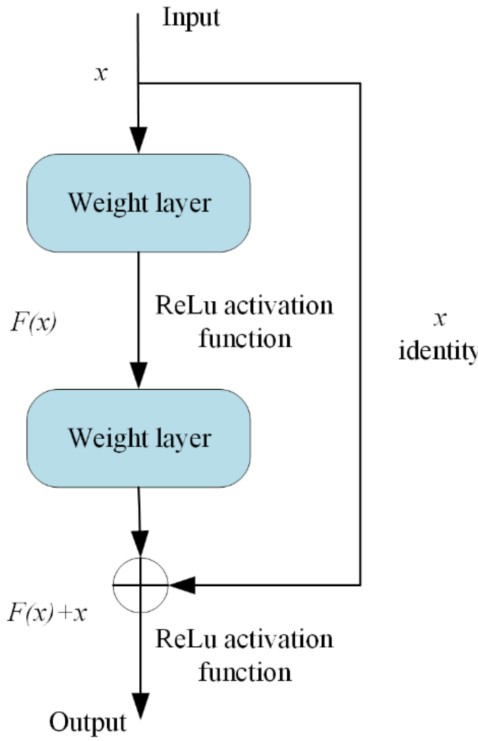

**Figure 3.** Residual block structure of ResNet.

In Figure 3, $x$ is the input, $F(x)$ is the residual mapping, and the rectified linear unit (ReLu) [23] is the activation function. The ReLu activation function is defined as

$$f(x) = \begin{cases} x, & x \geq 0 \\ 0, & \text{otherwise} \end{cases}.$$ 

(4)

In this study, we adopt ResNet50, whose residual network depth is 50. It is composed of multiple residual blocks. Figure 4 depicts its overall structure.

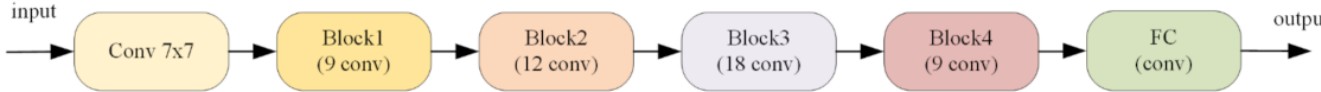

**Figure 4.** Residual block structure of ResNet50.

2.2.2. Faster R-CNN

Figure 5 illustrates the basic network framework of the Faster R-CNN. Different from other R-CNN networks, the RPN is introduced for the first time herein. Detection boxes are generated directly by the RPN. GPU parallel computing is used to greatly accelerate the whole network process.

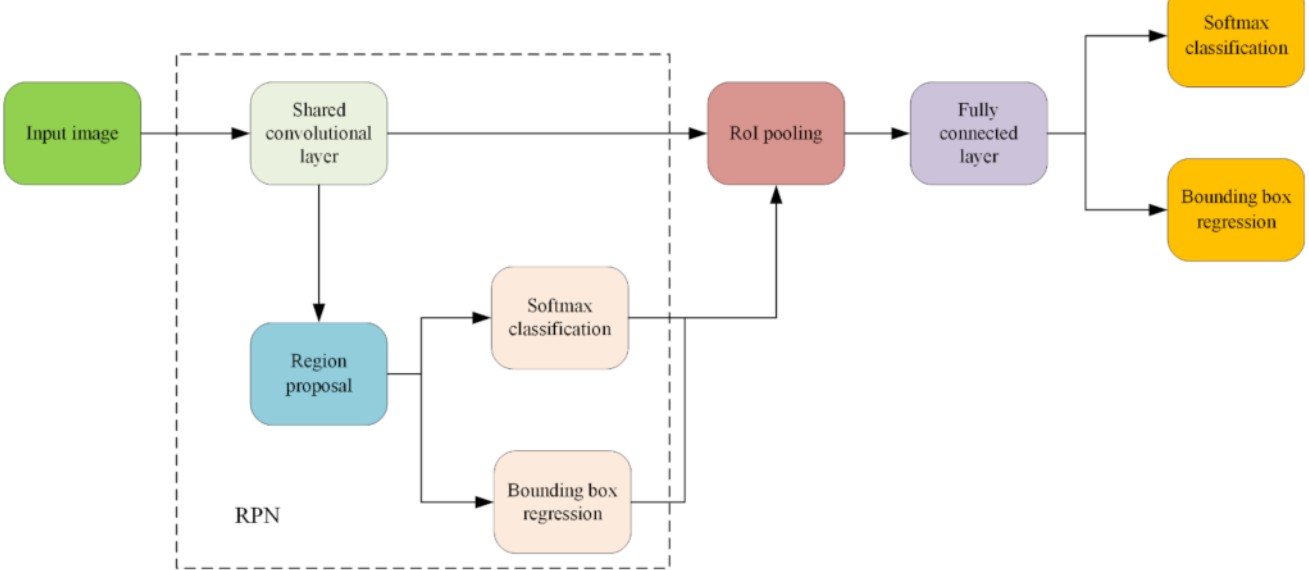

**Figure 5.** Basic network framework of the Faster R-CNN.

When an image is input, the feature map is first obtained by the backbone feature extraction network. The feature map is then sent to the RPN to obtain the preliminary region proposals. The region proposals are obtained by sliding an $N \times N$ convolution kernel over the convolutional feature map with a certain step size. Nine default reference bounding boxes can be found at the center of each slide window. These reference boxes are usually known as "anchors". The convolutional results are mapped into feature vectors, which are then input into two separate fully connected layers for the RPN classification and regression. The classification layer uses the SoftMax [24] function to calculate the likelihood probability of the output categories. Its outputs have $2 \times 9$ values used to judge whether the anchors are a target or a background. The regression layer outputs $4 \times 9$ values representing the coordinates of the center points of the nine anchors and the fine-tuning information of the width and the height.

The size of the anchor boxes is related to the aspect ratio and scale set by a human. The default aspect ratios are 1:1, 1:2, and 2:1, and the scales are $128^2$, $256^2$, and $512^2$. It can be combined into nine kinds of anchors, as shown in Figure 6. The red, green, and blue rectangles represent three aspect ratios for each scale.

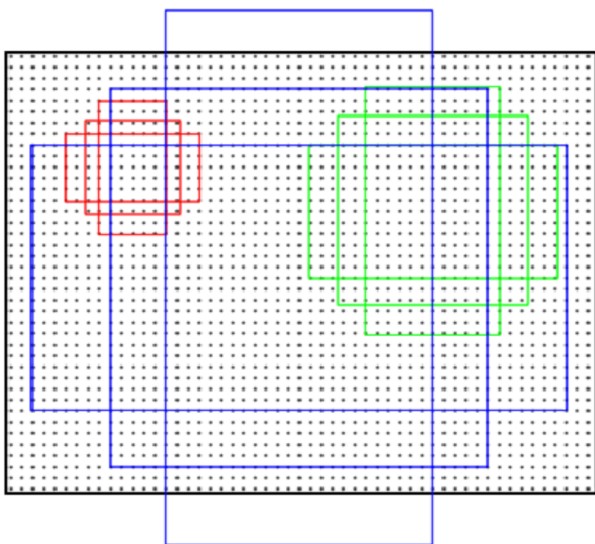

**Figure 6.** Diagram of the nine kinds of anchors.

Figure 7 shows the RPN structure. Positive and negative samples are introduced and represented by the truth value in the RPN prediction stage. The positive sample refers to the part where the overlap rate of the anchor and the ground truth exceeds 0.7. The negative sample refers to the part where the overlap rate is less than 0.3, and the others are ignored. After the sample is divided, the total loss of the RPN is calculated by the multitask loss function. The calculation formula is

$$L(\{p_i\}, \{t_i\}) = \frac{1}{N_{cls}} \sum_i L_{cls}(p_i, p_i^*) + \lambda \frac{1}{N_{reg}} \sum_i p_i^* L_{reg}(t_i, t_i^*), \tag{5}$$

where $N_{cls}$ represents the number of batch training data, $N_{reg}$ represents the number of anchors, and $\lambda$ represents the balance weight. $L_{cls}(p_i, p_i^*)$ is the logarithmic loss function defined as

$$L_{cls}(p_i, p_i^*) = -\log[p_i^* p_i + (1 - p_i^*)(1 - p_i)], \tag{6}$$

and $L_{reg}(t_i, t_i^*)$ is the regression loss calculated by the following *Smooth L*1 function:

$$L_{reg}(t_i, t_i^*) = \begin{cases} 0.5(t_i - t_i^*)^2, & |x| < 1 \\ |t_i - t_i^*| - 0.5, & \text{otherwise} \end{cases}, \tag{7}$$

where $p_i$ is the probability of the anchor being predicted as the target, and $p_i^*$ is the truth value of the prediction result: if the anchor is predicted as a positive sample, the value of tag $p_i^*$ is 1; otherwise, the value is 0; $t_i = \{t_x, t_y, t_w, t_h\}$ is the location of the predicted detection box; and $t_i^*$ is the ground truth coordinate.

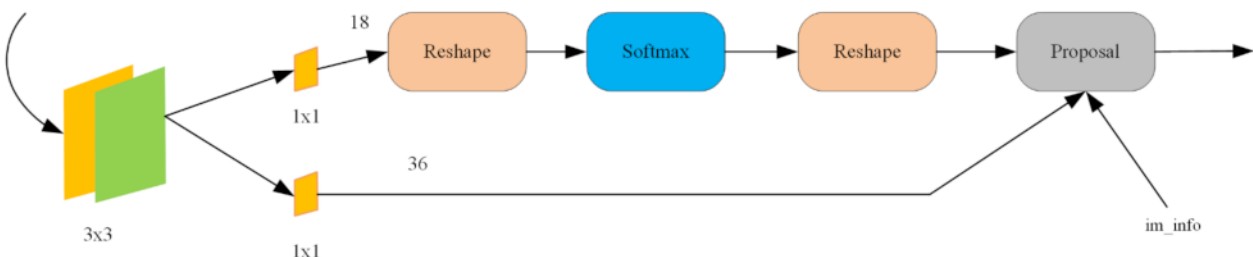

**Figure 7.** RPN structure.

### 2.3. Modified Faster R-CNN Structure

On the basis of the original Faster R-CNN target detection method, some modifications have been made to improve the performance of the detector, such as using VGG16 to replace the original backbone feature extraction network, introducing the K-means clustering algorithm to initialize anchors, and exploiting soft non-maximum suppression to eliminate the overlapped detection boxes.

### 2.3.1. VGG16 Feature Extraction Network

The VGG16 feature extraction network was proposed by the Oxford University Computer Vision Group in 2014. It builds a CNN with 16 layers by stacking a $3 \times 3$ convolution kernel and a $2 \times 2$ maximum pooling layer. The network consists of thirteen convolution layers, five pooling layers, and three fully connected layers. Among them, the total number of the convolution and fully connected layers with a weight coefficient is 16; hence, it is named VGG16. Figure 8 displays the VGG16 structure.

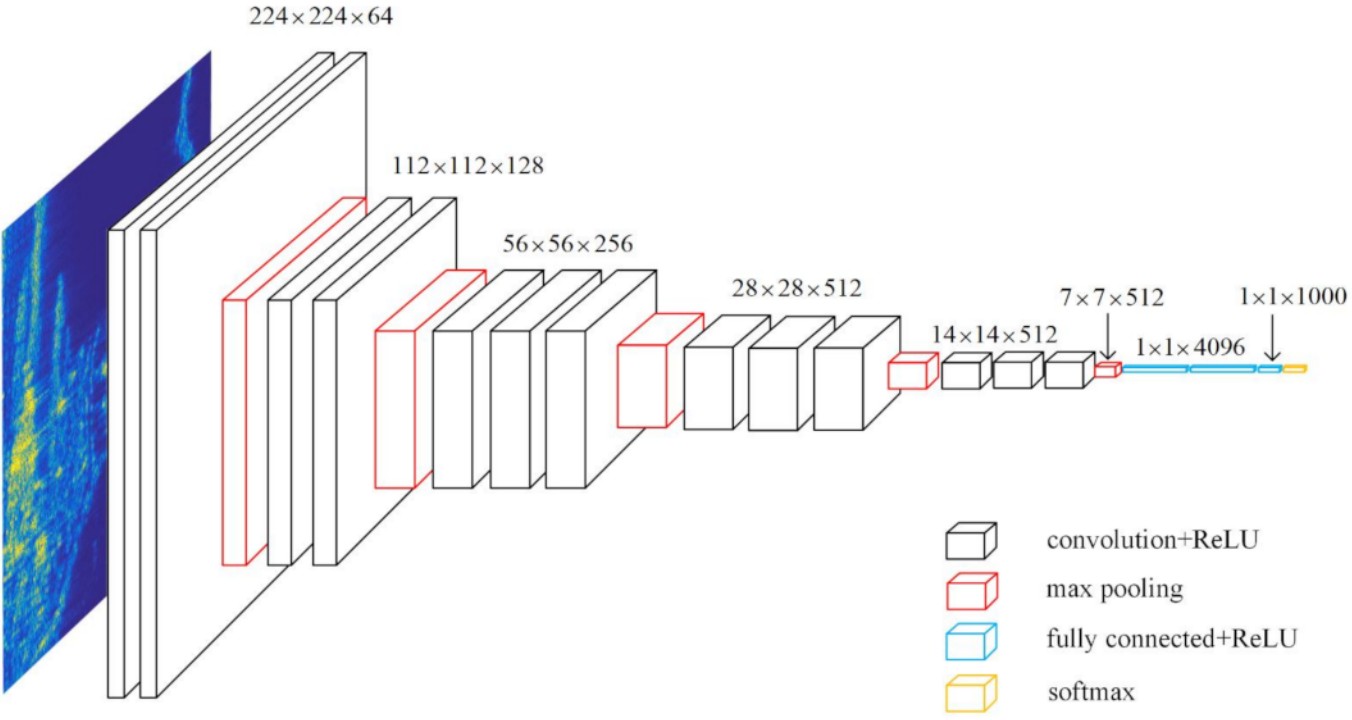

**Figure 8.** VGG16 structure.

The convolution kernel size of each convolution layer in VGG16 is $3 \times 3$. The size of the obtained feature map remains unchanged by convoluting the image with one step and one padding. The same convolution kernel size is used because two $3 \times 3$ convolution kernels in a series are equivalent to a $5 \times 5$ convolution kernel, and their receptive fields are the same, but two $3 \times 3$ convolution kernels need fewer parameters. In addition, stacking the small convolution kernels many times increases the times of using the activation function and enhances the abstract ability of the feature extraction. The last three layers are fully connected layers. The first two layers obtain 4096 dimensional vectors. The last layer obtains 1000 dimensional vectors corresponding to 1000 classification results. The ReLU activation function is used in all the convolutional and fully connected layers of the entire network.

According to the characteristics of the data set used in the experiment, we consider using the relatively shallow VGG16 feature extraction network because of the small size of the target, low resolution in the azimuth cell, and the low-level feature layer containing more location information and detailed information. We also replace the ReLU activation

function with the PReLU [25], which can adaptively learn the rectifier parameters and improve model fitting with nearly zero extra computational cost and small overfitting risk. The PReLU activation function is defined as

$$f(x_i) = \begin{cases} x_i, & x_i > 0 \\ a_i x_i, & \text{otherwise} \end{cases},$$

(8)

where $x_i$ is the input of the nonlinear activation on the $i$th channel and $a_i$ is the coefficient controlling the slope of the negative part. Subscript $i$ indicates that the nonlinear activation varies on different channels. When $a_i = 0$, it becomes ReLU. When $a_i = 0.01$, it becomes Leaky ReLU [25]. When $a_i$ is a learnable parameter, Equation (8) is named PReLU.

We adopt the PReLU activation function to improve the network accuracy. The PReLU parameters are learned adaptively together with the whole model in the end-to-end training, which can lead to more specialized activations. Backpropagation [26] is used to train PReLU, and PReLU can be optimized with other layers at the same time. The update formulations of $\{a_i\}$ are derived from the chain rule. The gradient of $a_i$ for one layer is

$$\frac{\partial \varepsilon}{\partial a_i} = \sum_{x_i} \frac{\partial \varepsilon}{\partial f(x_i)} \frac{\partial f(x_i)}{\partial a_i},$$

(9)

where $\varepsilon$ represents the objective function. The term $\frac{\partial \varepsilon}{\partial f(x_i)}$ is the gradient propagated from the deeper layer. The summation $\sum_{x_i}$ runs over all positions of the feature map. We adopt the momentum method when updating $a_i$:

$$\Delta a_i := \mu \Delta a_i + \epsilon \frac{\partial \varepsilon}{\partial a_i}.$$

(10)

Here, $\mu$ is the momentum and $\epsilon$ is the learning rate. We do not use the weight decay when updating $a_i$ to prevent the PReLU from biasing toward the ReLU. Furthermore, we do not constrain the range of $a_i$; hence, the activation function may be non-monotonic. The initial value of $a_i$ is set to 0.25.

The following experiments show that the classification accuracy can be improved by replacing the parameter-free ReLU activation with a learned parameter activation unit.

2.3.2. K-Means Clustering Algorithm Initializes Anchors

The anchor is an important parameter of the Faster R-CNN target detector. The anchor shape and quantity will affect the detector's efficiency and accuracy. The default anchor scales in the RPN are $128^2$, $256^2$, and $512^2$, and the default aspect ratios are 1:1, 1:2, and 2:1. This can be combined into nine kinds of anchors. However, for different data sets, the default size of anchors cannot meet the real situation. Improper anchors will not only slow down the network convergence speed but also affect the error function calculation. This section abandons the traditional Faster R-CNN strategy of manually setting the anchor and introduces the K-means clustering algorithm to set anchors.

Figure 9 shows the real situation of all ground truths in the data set. Most of the ground truths have similar sizes and shapes. In general, the distribution of ground truths is scattered. Therefore, it is very difficult to determine the appropriate anchor manually. However, by using the clustering algorithm, we set a clustering number and the similar ground truths can be divided together, in which each division represents a cluster, and each cluster center can be set as an anchor. The distance measurement formula used is

$$d(\text{box}, \text{cent}) = 1 - IoU(\text{box}, \text{cent}),$$

(11)

where box represents the ground truth coordinate: $(x_i, y_i, w_i, h_i), i \in \{1, 2, \cdots, k\}$; $k$ represents the total number of samples; cent represents the cluster center: $(X_n, Y_n, W_n, H_n)$,

$n \in \{1, 2, \cdots, N\}$; $N$ represents the number of clusters; and $IoU(\text{box}, \text{cent})$ represents the intersection over union (IoU) of the ground truth and the cluster center.

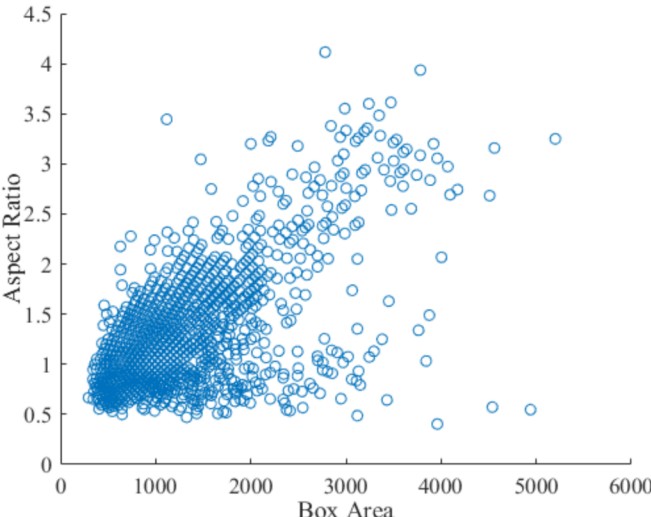

**Figure 9.** Box area vs. aspect ratio.

Different from the Euclidean distance measurement, the distance measurement based on the IoU does not change with the box size. The Euclidean distance measurement will produce a larger error with the increase in the box size. In addition, using the IoU distance measurement brings together boxes with a similar aspect ratio and size, which is more suitable for the anchor estimation of the data set. However, the following problem is to determine the number of clusters; that is, the number of anchors. We usually use empirical analysis, but herein, we propose an idea of traversal, setting different cluster numbers $N$ and calculating the mean IoU of the ground truths and the cluster center under each cluster number.

If the mean IoU value is greater than 0.5, the anchor and the bounding box (i.e., ground truth) in the data set can be well overlapped. Increasing the number of anchors can improve the measurement value of the mean IoU. However, using more anchors increases the calculation cost and leads to overfitting, which will eventually lead to the degradation of the detector performance. Figures 9 and 10 show that for the given data set, the number of nine anchors in the original Faster R-CNN seems reasonable; however, the aspect ratio and the area cannot fully meet the actual needs. The anchor location is uncertain; thus, only the width and the height are used for the calculation. The eight anchors obtained by the K-means clustering algorithm are (55,44), (30,25), (31,53), (37,35), (34,87), (28,37), (25,29), and (25,21). Each anchor has the format (*height, width*). The anchors are in accordance with the target characteristics and verify the effectiveness of the K-means clustering algorithm.

### 2.3.3. Soft-NMS

In the final detection results, the same ship target may be contained by multiple target detection boxes. When the distance between the targets is too close, the NMS algorithm [27] will rudely delete the overlapping detection boxes higher than the IoU threshold, leading to a missed target detection. Therefore, we use the Soft-NMS algorithm to extract the most suitable target detection box using the following formula:

$$s_i = \begin{cases} s_i, & IoU(M, b_i) < N_t \\ s_i \times e^{-\frac{IoU(M,b_i)^2}{0.5}}, & IoU(M, b_i) \geq N_t \end{cases}, \tag{12}$$

where $s_i$ represents the score of the current category detection box, $M$ represents the detection box corresponding to the highest score, $b_i$ represents the current detection box, $IoU(M, b_i)$ represents their IoU, and $N_t$ represents the Soft-NMS threshold.

The biggest difference between Soft-NMS and NMS is that the former uses the weight penalty strategy to reduce the confidence of the detection box with an IoU higher than the threshold. In the algorithm, we use the Gaussian weighting function, which is more continuous and smoother than the linear weighting function in penalty. The penalty increases when the detection box $b_i$ and $M$ have a high overlap, resulting in a serious decline in confidence. Finally, we set a reasonable confidence threshold to decide whether or not to delete the suppressed detection box. Algorithm 1 describes the Soft-NMS process.

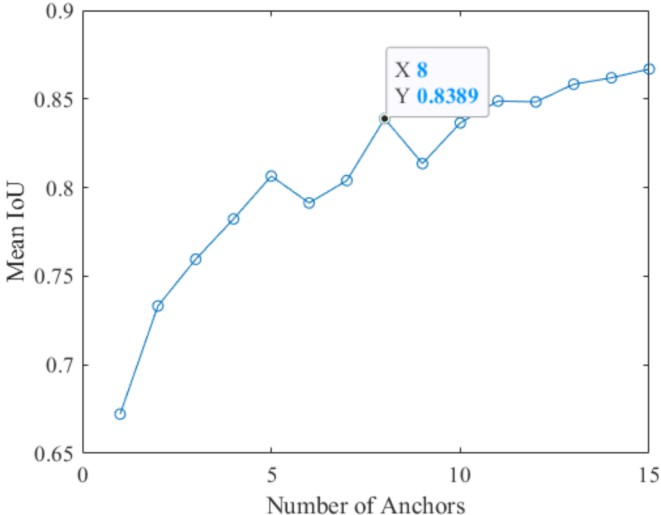

**Figure 10.** Number of anchors vs. the mean IoU.

| **Algorithm 1.** Soft-NMS |
|---|
| **Input**: The list of initial detection boxes $B = \{b_1, b_2, \ldots, b_N\}$, and the corresponding detection scores $S = \{s_1, s_2, \ldots, s_N\}$ |
| 1:   **Set**: $D = \varnothing$ |
| 2:   **while** $B \neq \varnothing$ **do** |
| 3:     $m = \text{argmax} S$ |
| 4:     $M = b_m$ |
| 5:     $D = D \cup M; B = B - M$ |
| 6:   **for** $b_i$ in $B$ **do** |
| 7:     $s_i = s_i f(iou(M, b_i))$ |
| 8:   **end for** |
| 9:   **end while** |
| **Output**: $D, S$ |

## 3. Experiments

This section introduces the processing steps of the proposed algorithm, experimental data set and environment configuration. Furthermore, the experimental results of the proposed algorithm and the CA-CFAR, GOCA-CFAR, and SOCA-CFAR algorithms are detailed presented and compared.

### 3.1. Target Detection Steps in a Coastal Defense Radar Image

Figure 11 illustrates the algorithm flow of the target detection of the coastal defense radar based on the Faster R-CNN. The specific steps are presented below:

(1) Preprocess the radar echo signals and transform the echo data into a pulse-range two-dimensional (2D) image for subsequent network training and testing.

(2) Make the training set label and the test set division. For the pulse-range 2D echo data image, the echo image is segmented according to the pulse number and under the condition that the target does not cross the range unit, and the aspect ratio of the image is close. They are cooperative targets; thus, their bounding boxes are determined according to the target position information provided by the GPS. We then use the annotation software, LabelImg, to complete the labeling work on the segmented amplitude image.

(3) Initialize the basic network parameters and use the K-means clustering algorithm to set the anchor size.

(4) Start the CNN training and use the gradient descent algorithm to calculate the error between the output of the network and the real target, then backpropagate the error to adjust the network parameters, such as the weight and the bias. Continue the training until the network converges or reaches the preset training times.

(5) Verify whether or not the network is underfitting or overfitting during the training process using the verification set of each epoch and adjusting the network parameters to continue training.

(6) After training, obtain the target detector of the coastal defense radar based on the Faster R-CNN. The Soft-NMS algorithm is added to eliminate the overlapping detection boxes and test the sample set. The target is detected from the radar amplitude image. We calculate the recognition accuracy rate, false-alarm rate, recall rate, and mean average precision (mAP) to determine whether or not the evaluation index is met. If it is satisfied, the target detector has completed the training and can be used for the target detection in the unknown sea-surface radar image; otherwise, return to step 3 to adjust the network parameters for retraining.

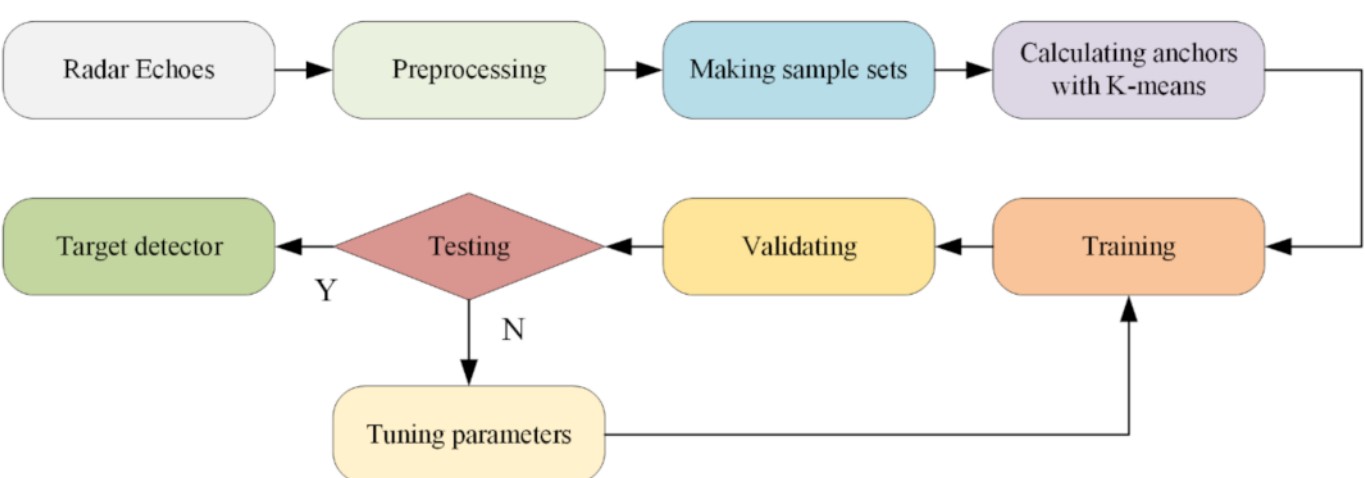

**Figure 11.** Flowchart of the target detection of the coastal defense radar based on the Faster R-CNN.

### 3.2. Build Data Set

The data in this section are the measured data obtained from the actual sea-surface detection by a certain organization using the coastal defense scanning radar. Figure 12 shows the result after the pulse data compression. The sea area has many islands, reefs, sea clutter, and interference. The target detection technology based on deep learning is dependent on the total number of samples in the data set. The target information contained

in the data set and the accuracy of the bounding box will also greatly affect the final trained detector; hence, the original measured data are enhanced by changing the aspect ratio and the contrast of the image, flipping and varying the scaling degrees. Finally, we obtain 1782 image samples and divide them into three groups according to the ratio of training, validation, and test sets of 8:1:1. The training set contains 1426 image samples, the verification set includes 178 image samples, and the test set contains 178 image samples. Each image group is balanced and contains a complete sea scene. The data set is then built.

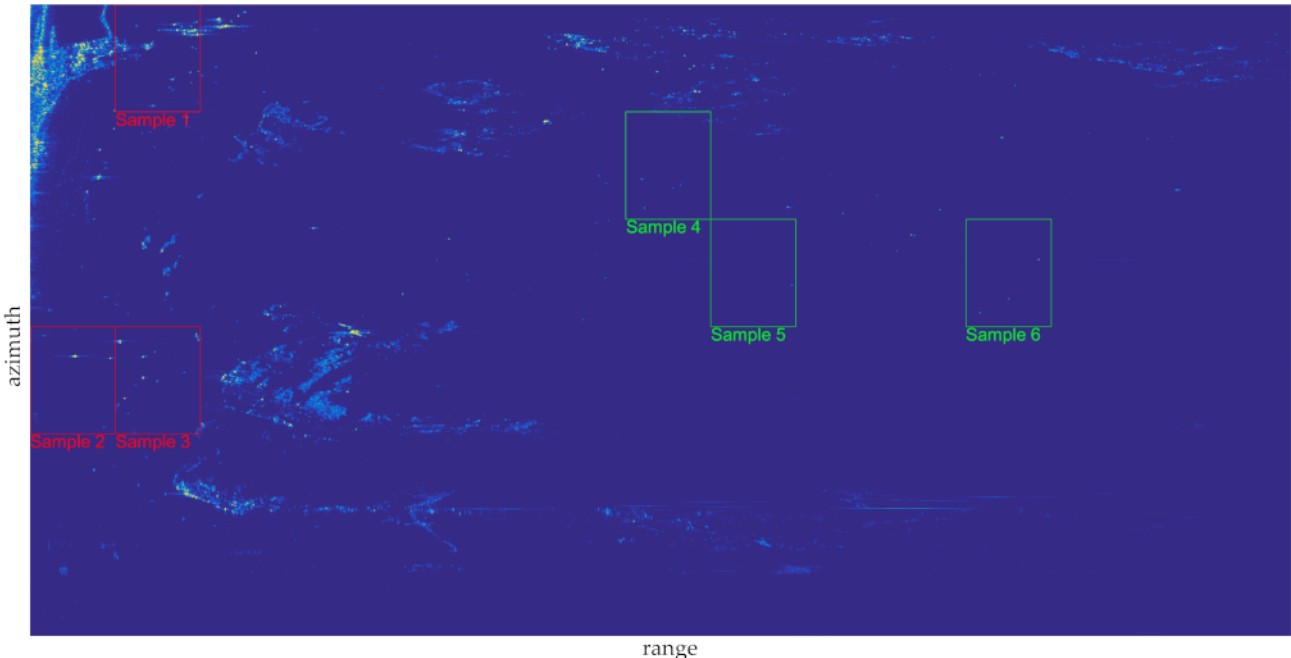

**Figure 12.** Measured data.

After the sea-surface radar image is segmented, we select six image samples in the offshore and distant water containing ship targets for the analysis. In Figure 12, the red rectangle indicates the target in the offshore water, whereas the green rectangle indicates the target in the distant water. We enlarge and display these areas to clearly observe the distribution of the ship targets, islands, reefs, sea clutter, and interference (Figures 13 and 14).

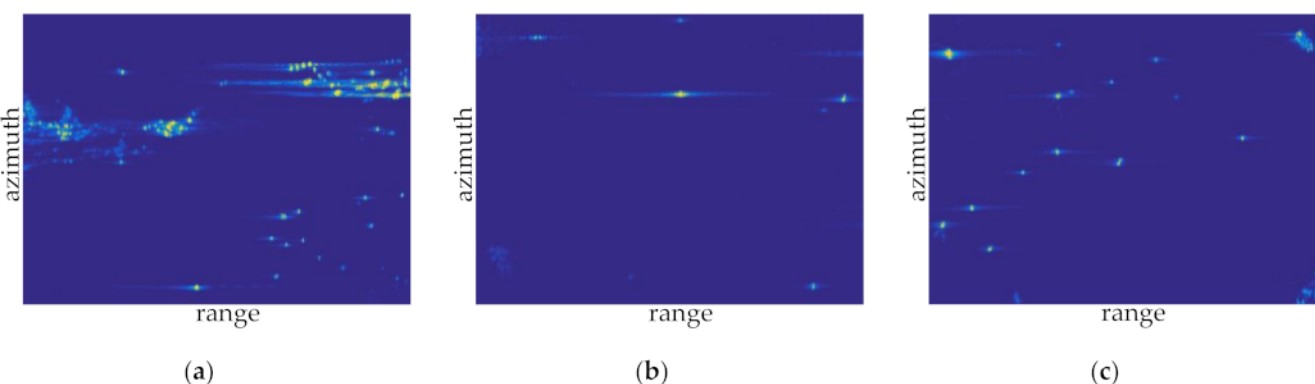

**Figure 13.** Image samples in offshore water: (**a**) sample 1; (**b**) sample 2; and (**c**) sample 3.

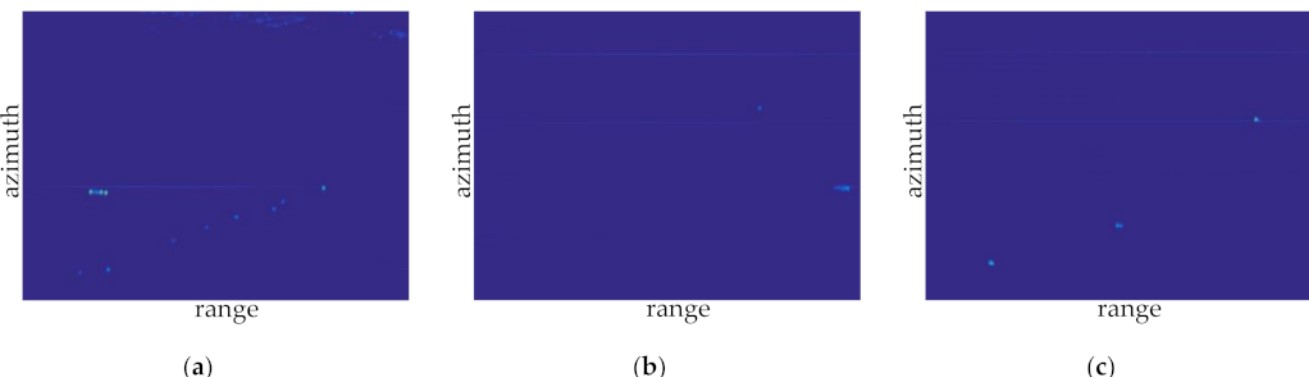

**Figure 14.** Image samples in distant water: (**a**) sample 4; (**b**) sample 5; and (**c**) sample 6.

In the process of making the training set, two helpful information are applied effectively to confirm the presence of the moving targets in the observation scene. The first information is the AIS (Automatic Identification System) [28] information provided by the ships in the observation scene. Additionally, the second information is the comparison of the radar echoes of the same observation scene at different time (the time span can be several days), since its highly unlikely that a ship would remain in the same place for several days. After the confirmation of the moving targets in the observation scene, the annotation software, which is named LabelImg, are used to complete the labeling work on the segmented amplitude radar image. The aforementioned six image samples are annotated, as shown in Figure 15.

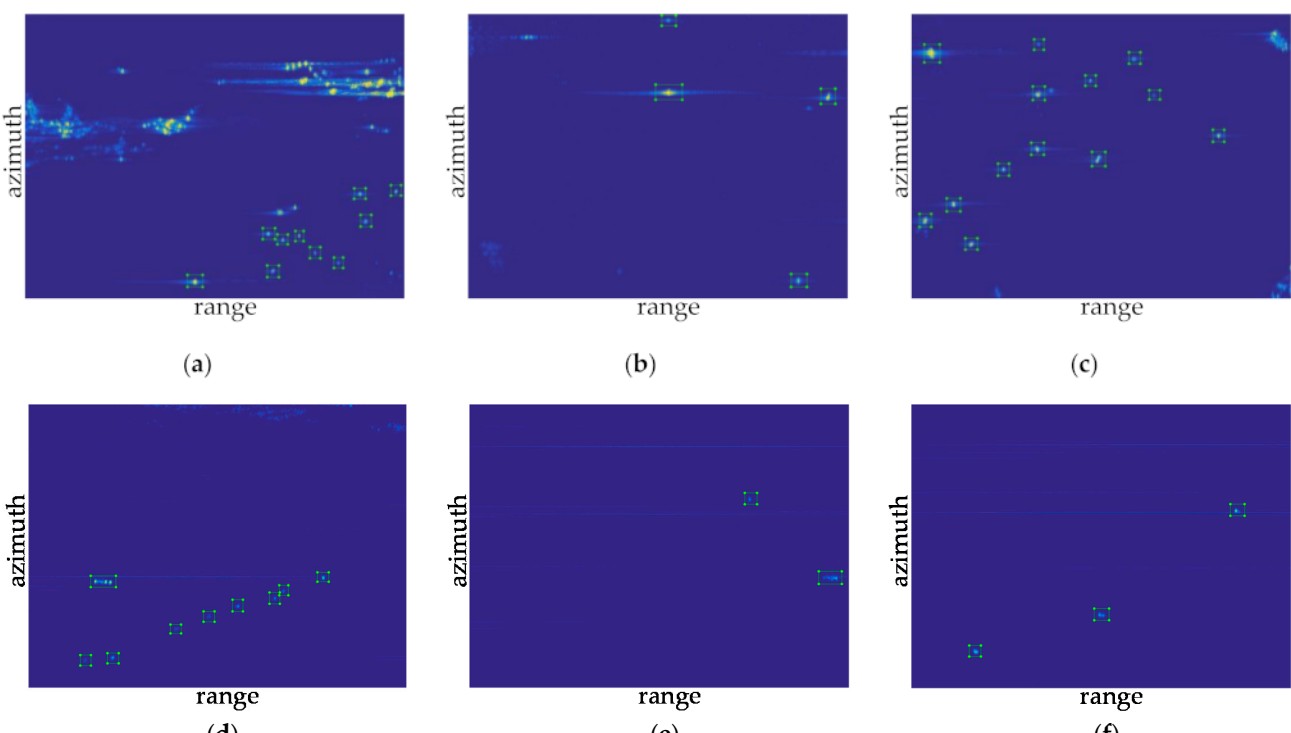

**Figure 15.** The annotated image samples: (**a**) sample 1; (**b**) sample 2; (**c**) sample 3; (**d**) sample 4; (**e**) sample 5; and (**f**) sample 6.

Figure 15a–c show the annotated results in offshore water. Additionally, there are 10 real ship targets in sample 1, four real ship targets in sample 2, and 13 real ship targets in sample 3, while for the distant water, since it is far from the coastline, the influence of islands and reefs is little. Figure 15d–f represent the annotated results in distant water,

where there are nine real targets in sample 4, two real targets in sample 5, and three real targets in sample 6.

### 3.3. Experimental Environment and Parameters

Table 1 shows the experimental hardware and software configuration. On this experimental platform, the Faster R-CNN is used for training and testing according to the algorithm flow in Figure 11.

**Table 1.** Experimental environment.

| System | Windows 10 | Tool | Anaconda 3 |
|---|---|---|---|
| **RAM** | 32 GB | **Programming** | Python 3.6 |
| **CPU** | Intel i5-9600KF @3.7 GHz $\times$ 6 | **IDE** | VS code |
| **GPU** | NVIDIA GTX 1080Ti 11 G | **Framework** | Pytorch-GPU |
| **Auxiliary tools** | MATLAB | **Others** | CUDA 10.0 |

For the parameters selection in Faster R-CNN algorithm, we adopt the Pytorch framework, selecting ResNet50 as the backbone feature extraction network, without using the K-means clustering algorithm to set the anchors. The Soft-NMS algorithm is used to complete the filter of the test results and training's iteration number of each epoch is 2000, with a total of 70 epochs. The initial learning rate (Lr) is set as 0.001, and when the iterations reach 70,000, it will be dropped to 0.0001. Additionally, the Adam optimizer is adopted, and the size of input image demands $600 \times 580$. The detection threshold value, that is, the confidence, is set as 0.5 to test the training results.

For the parameters selection in the CFAR algorithm, after the consideration of the range and azimuth resolution of our radar system, the numbers of protection band cells in the row and column direction are set to 2 and 30, respectively, and the numbers of reference band cells are both set to 1. Besides, the number of reference cells is set to 136 and the false alarm rate is $10^{-7}$.

### 3.4. Evaluation Criteria and Results

The final detection result is evaluated by the recognition accuracy rate, false alarm number, recall rate, and mAP. The detection time of the Faster R-CNN and three CFAR methods is also counted. We define the recognition accuracy rate $P_{ra}$ as

$$P_{ra} = \frac{N_{td}}{N_{td} + N_{fd}}, \tag{13}$$

where $N_{td}$ represents the number of the real targets predicted correctly and $N_{fd}$ represents the number of false targets predicted as correct targets, i.e., the false alarm number. The recall rate $R$ is defined as

$$R = \frac{N_{td}}{N_{gt}}, \tag{14}$$

where $N_{gt}$ represents the total number of ground truths.

Figures 16 and 17 illustrate the detection results of the samples for the CFAR methods, comparing the detection performance of the CA-CFAR, GOCA-CFAR, and SOCA-CFAR algorithms.

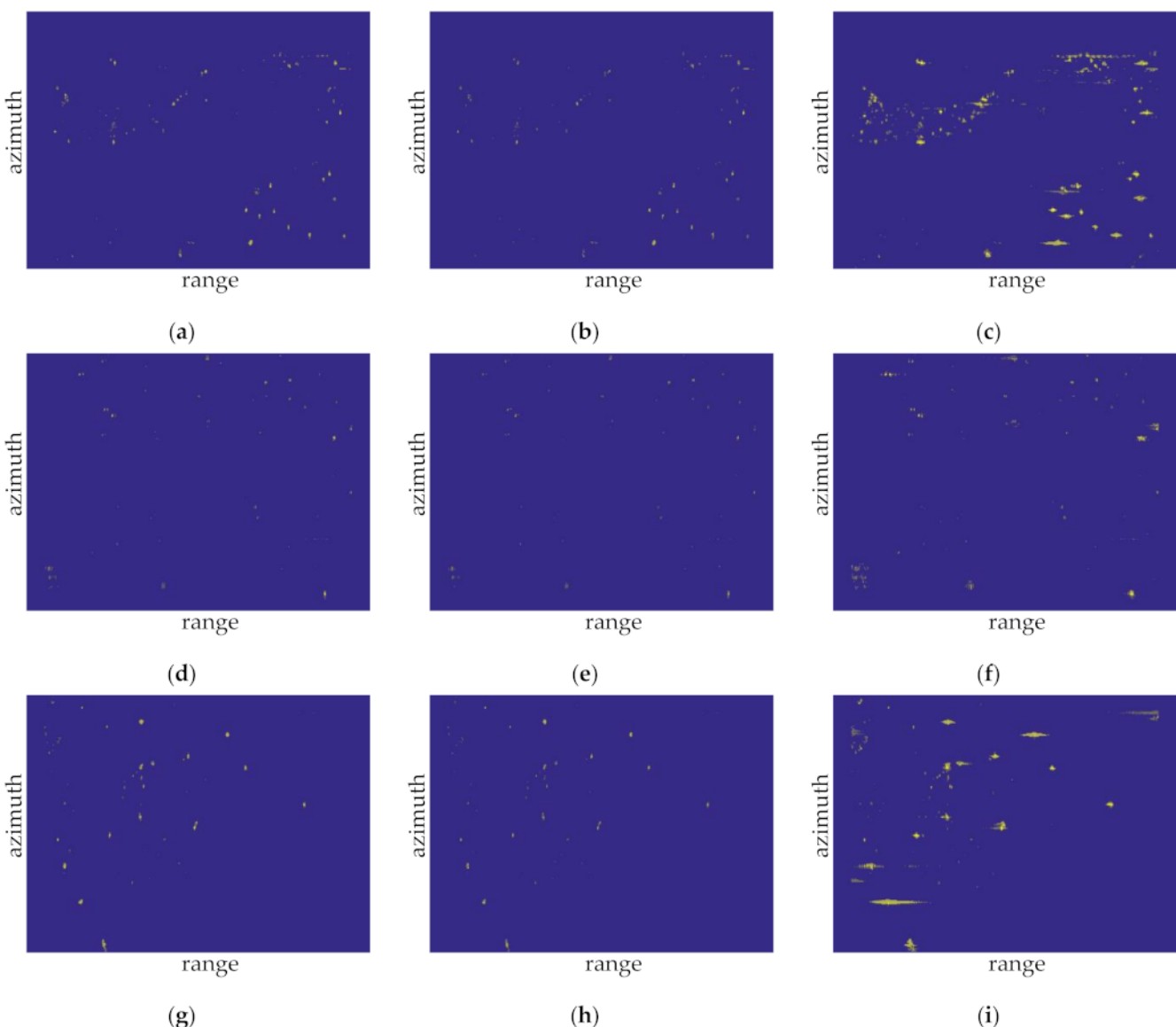

**Figure 16.** Detection results of the samples for the CA-CFAR, GOCA-CFAR, and SOCA-CFAR algorithms in offshore water: (**a**) the CA-CFAR detection results of sample 1; (**b**) the GOCA-CFAR detection results of sample 1; (**c**) the SOCA-CFAR detection results of sample 1; (**d**) the CA-CFAR detection results of sample 2; (**e**) the GOCA-CFAR detection results of sample 2; (**f**) the SOCA-CFAR detection results of sample 2; (**g**) the CA-CFAR detection results of sample 3; (**h**) the GOCA-CFAR detection results of sample 3; and (**i**) the SOCA-CFAR detection results of sample 3.

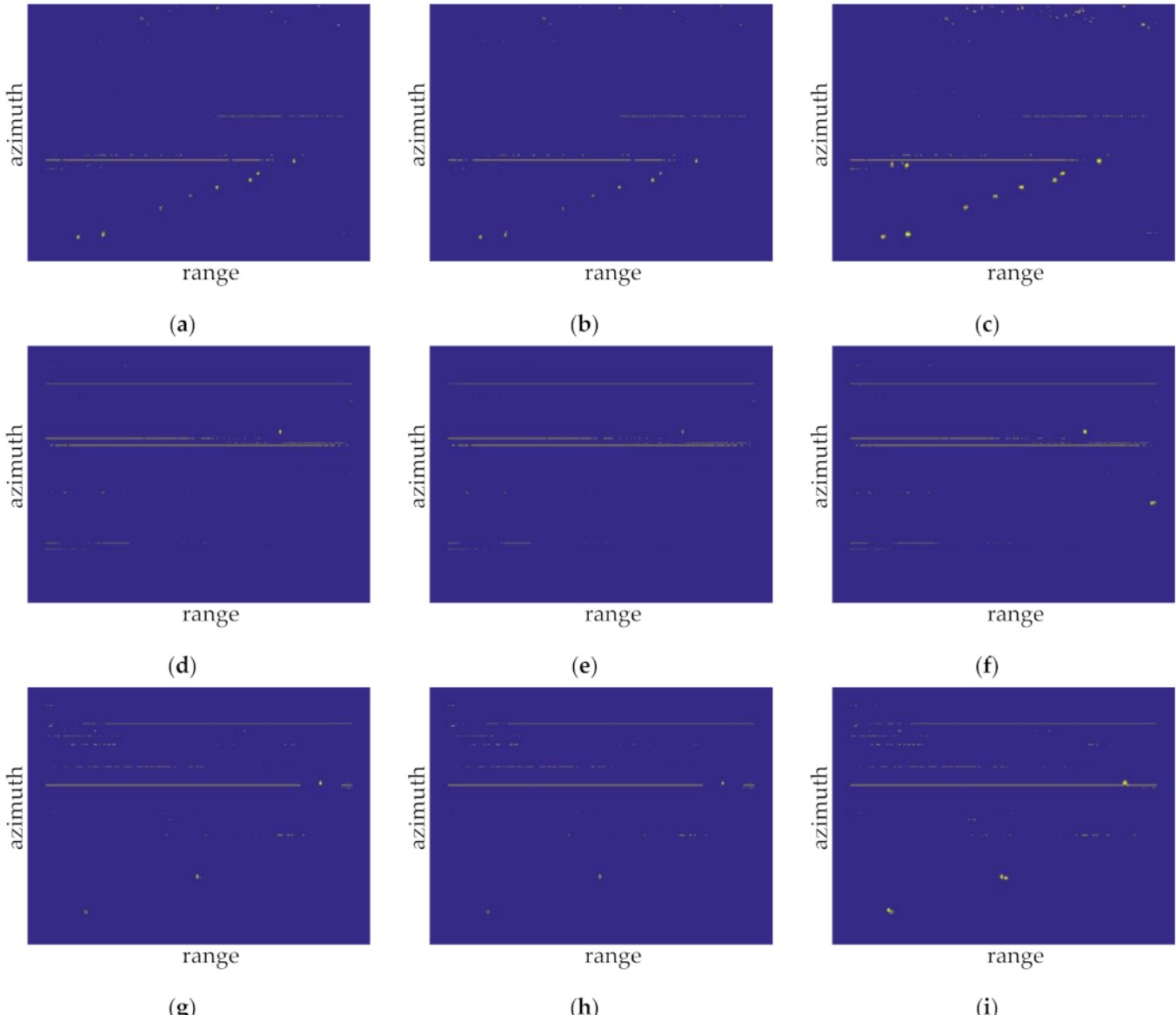

**Figure 17.** Detection results of the samples for the CA-CFAR, GOCA-CFAR, and SOCA-CFAR algorithms in distant water: (**a**) the CA-CFAR detection results of sample 4; (**b**) the GOCA-CFAR detection results of sample 4; (**c**) the SOCA-CFAR detection results of sample 4; (**d**) the CA-CFAR detection results of sample 5; (**e**) the GOCA-CFAR detection results of sample 5; (**f**) the SOCA-CFAR detection results of sample 5; (**g**) the CA-CFAR detection results of sample 6; (**h**) the GOCA-CFAR detection results of sample 6; and (**i**) the SOCA-CFAR detection results of sample 6.

The processing results of CFAR algorithm are described as follows.

Sample 1 is close to the coastline; thus, islands and reefs can be found in the scene, except for the targets located in the upper part of the image. Figure 16a–c illustrate the detection results of three CFAR methods. All of the three CFAR methods detect nine targets correctly, one target is missed. Many false alarms appear in the islands and reefs region in the upper part of the image, and the number of false alarms is 255, 207, and 699, respectively. Therefore, according to Equations (13) and (14), the recognition accuracy rates of three CFAR methods are 3.40%, 4.16%, and 1.27%, respectively, and the recall rates are all 90%.

Meanwhile, sample 2 shows a strong sea clutter in the lower left corner of the image and a reef in the upper left corner. Figure 16d–f show the CFAR detection results. All the three CFAR methods detect four targets correctly; no target is missed. The number of false

alarms is 155, 129, and 339, respectively. The recognition accuracy rates of three CFAR methods are 2.51%, 3.00%, and 1.16%, respectively, and the recall rates are all 100%.

Sample 3 depicts an island in the upper and lower right corners of the image. Figure 16g–i show the different CFAR detection results. The CA-CFAR and GOCA-CFAR detect 12 targets correctly, and one target is missed, while the SOCA-CFAR detects 13 targets correctly, and no target is missed. The false alarm number is 162, 143, and 385, respectively. Therefore, the recognition accuracy rates of three CFAR methods are 6.89%, 7.74%, and 3.26%, respectively, and the recall rates are 92.30%, 92.30%, and 100%, respectively.

Sample 4 exhibits a few reefs in the upper part of the image. Besides, interference component can be found in the middle of the image. Figure 17a–c present the CFAR detection results. All the three CFAR methods detect nine targets correctly, and no target is missed. The number of false alarms is 121, 103, and 181, respectively. Therefore, the recognition accuracy rates of three CFAR methods are 6.92%, 8.03%, and 4.73%, respectively, and the recall rates are all 100%.

Sample 5 shows the strong interference component that exists in the image. Figure 17d–f show the CFAR detection results. The CA-CFAR and GOCA-CFAR detect one target correctly, and one target is missed. The SOCA-CFAR detects two targets correctly, and no target is missed. The number of false alarms is 97, 83, and 99, respectively. The recognition accuracy rates of three CFAR methods can be calculated as 1.02%, 1.19%, and 1.98%, respectively, and the recall rates are 50%, 50%, and 100%, respectively.

Sample 6 also shows the strong interference component in the image. The CFAR detection results are shown in Figure 17g–i. All the CFAR methods detect three targets correctly, and no target is missed. The number of false alarms is 114, 99, and 116, respectively. The recognition accuracy rates of three CFAR methods are 2.56%, 2.94%, and 2.52%, respectively, and the recall rates are all 100%.

Based on aforementioned results, Table 2 can be obtained.

**Table 2.** Statistics of the detection results of the samples for the CA-CFAR, GOCA-CFAR, and SOCA-CFAR algorithms.

| | Detection Algorithm | $N_{td}$ | $N_{fd}$ | $N_{gt}$ | $P_{ra}$/% | $R$/% | $T$/s |
|---|---|---|---|---|---|---|---|
| **Sample 1** | CA-CFAR | 9 | 255 | 10 | 3.40 | 90 | 0.9569 |
| | GOCA-CFAR | 9 | 207 | 10 | 4.16 | 90 | 1.3561 |
| | SOCA-CFAR | 9 | 699 | 10 | 1.27 | 90 | 1.3853 |
| **Sample 2** | CA-CFAR | 4 | 155 | 4 | 2.51 | 100 | 0.9594 |
| | GOCA-CFAR | 4 | 129 | 4 | 3.00 | 100 | 1.3583 |
| | SOCA-CFAR | 4 | 339 | 4 | 1.16 | 100 | 1.3347 |
| **Sample 3** | CA-CFAR | 12 | 162 | 13 | 6.89 | 92.30 | 0.9752 |
| | GOCA-CFAR | 12 | 143 | 13 | 7.74 | 92.30 | 1.3653 |
| | SOCA-CFAR | 13 | 385 | 13 | 3.26 | 100 | 1.3520 |
| **Sample 4** | CA-CFAR | 9 | 121 | 9 | 6.92 | 100 | 0.9582 |
| | GOCA-CFAR | 9 | 103 | 9 | 8.03 | 100 | 1.5329 |
| | SOCA-CFAR | 9 | 181 | 9 | 4.73 | 100 | 1.5749 |
| **Sample 5** | CA-CFAR | 1 | 97 | 2 | 1.02 | 50 | 0.9541 |
| | GOCA-CFAR | 1 | 83 | 2 | 1.19 | 50 | 1.3584 |
| | SOCA-CFAR | 2 | 99 | 2 | 1.98 | 100 | 1.3374 |
| **Sample 6** | CA-CFAR | 3 | 114 | 3 | 2.56 | 100 | 0.9534 |
| | GOCA-CFAR | 3 | 99 | 3 | 2.94 | 100 | 1.3754 |
| | SOCA-CFAR | 3 | 116 | 3 | 2.52 | 100 | 1.3626 |

The processing results of proposed algorithm are shown as follows.

For the offshore water sample 1 shown in Figure 18a, nine targets (marked with a red rectangle box) are detected correctly, one target is missed, and there is no false-alarm target. Therefore, the recognition accuracy rate is 100% and the recall rate is 90%. Figure 18b shows that the Faster R-CNN target detector detects four targets correctly, misses no targets, and

detects no false alarms. The recognition accuracy rate is 100% and the recall rate is 100%. In sample 3, 12 targets are detected correctly, one target is missed, and there is no false-alarm target. Therefore, the recognition accuracy rate is 100% and the recall rate is 92.31%. For image samples in distant water, the Faster R-CNN also shows superior performance. In Figure 18d–f, the Faster R-CNN detects all targets correctly and no false alarm of the samples. Therefore, the recognition accuracy rates are all 100% and the recall rates are all 100%. The processing results of the proposed algorithm are summarized in Table 3.

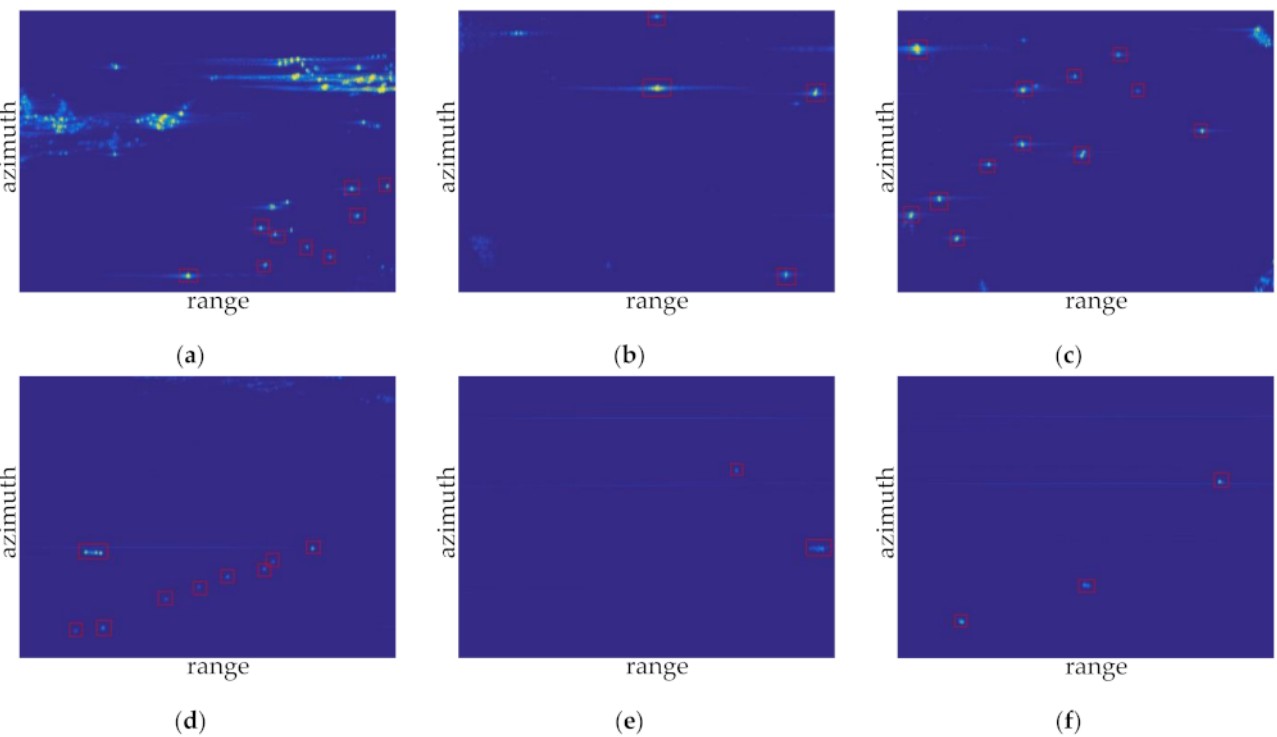

**Figure 18.** Detection results of the Faster R-CNN: (**a**) sample 1; (**b**) sample 2; (**c**) sample 3; (**d**) sample 4; (**e**) sample 5; and (**f**) sample 6.

**Table 3.** Statistics of the Faster R-CNN detection results.

|  | $N_{td}$ | $N_{fd}$ | $N_{gt}$ | $P_{ra}$/% | $R$/% | $T$/s |
|---|---|---|---|---|---|---|
| **Sample 1** | 9 | 0 | 10 | 100 | 90 | 0.6562 |
| **Sample 2** | 4 | 0 | 4 | 100 | 100 | 0.5983 |
| **Sample 3** | 12 | 0 | 13 | 100 | 92.31 | 0.6661 |
| **Sample 4** | 9 | 0 | 9 | 100 | 100 | 0.7041 |
| **Sample 5** | 2 | 0 | 2 | 100 | 100 | 0.6692 |
| **Sample 6** | 3 | 0 | 3 | 100 | 100 | 0.5734 |

### 3.5. Further Comparative Experiments

To reflect the influence of our improvement on the target detector based on the Faster R-CNN, we performed many comparative experiments on the basis of the above-mentioned experiments and analyzed the experimental results. The total number of ground truths in the test set was 394. The mAP and the average detection time of images were counted, given the threshold value of 0.5. Table 4 lists the specific experimental contents and results.

**Table 4.** Further experiments and results.

| Method | $N_{td}$ | $N_{fd}$ | $N_{gt}$ | $P_{ra}$/% | R/% | T/s | mAP/% |
|---|---|---|---|---|---|---|---|
| ResNet50 | 342 | 55 | 394 | 86.15 | 86.80 | 0.7042 | 81.41 |
| ResNet50 + K-means | 367 | 42 | 394 | 89.73 | 93.15 | 0.8317 | 87.20 |
| VGG16 | 320 | 22 | 394 | 93.57 | 81.22 | 0.2648 | 81.52 |
| VGG16 + K-means | 370 | 35 | 394 | 91.36 | 93.91 | 0.4559 | 91.42 |
| Improved Faster R-CNN | 370 | 32 | 394 | 92.04 | 93.91 | 0.3904 | 92.27 |

## 4. Discussion

Table 2 summarizes the detection results of all samples for the three CFAR algorithms. Whether in offshore water or distant water, both CA-CFAR and GOCA-CFAR can detect the same target number correctly. However, SOCA-CFAR detects more correct targets and many wrong targets at the same time, because SOCA-CFAR has a smaller detection threshold by calculating the reference cell mean for each half and then selecting the smallest mean. In terms of the detection time, the average detection time of CA-CFAR, GOCA-CFAR and SOCA-CFAR are 0.9595s, 1.3910s and 1.3911s, respectively. Therefore, CA-CFAR has the advantage of the fast detection time and GOCA-CFAR has the advantage of the high recognition accuracy rate. Taking that into consideration, we can draw a conclusion that CA-CFAR and GOCA-CFAR have a better performance than the SOCA-CFAR algorithm for the target detection of coastal defense radar according to the results in our measured data. It is up to the researchers to adopt which algorithm on the practical application.

Although in offshore water, samples 1 and 3 have a missed detection because of the weak echo energy of some targets, compared with the high false alarm number of the CFAR detector, the Faster R-CNN will not cause a false alarm. In distant water, the Faster R-CNN can avoid false detection and detect ship targets accurately as there are fewer islands and reefs with a strong echo energy. In addition, strong sea clutter and interference were observed, which made the CA-CFAR detector cause more false alarms. The number of false alarms was higher than that of the Faster R-CNN. In sample 5, the CA-CFAR and GOCA-CFAR detectors have a missed detection, which made the final recall rate lower than that of the Faster R-CNN. In terms of the detection time, the average detection time of the Faster R-CNN was 0.6489 s, whereas that of the CA-CFAR was 0.9595 s and the GOCA-CFAR was 1.3910 s. The detection time of the Faster R-CNN is faster than that of them.

Table 4 lists the results of specific further comparative experiments. We compared the influences of the VGG16 and ResNet50 feature extraction networks on the target detector and added the K-means clustering algorithm to calculate the anchor size. The detection result of the VGG16 network was better than that of ResNet50 for this data set. The fundamental reason is that the target size in the data set was small, and the resolution was lower than that of the ordinary optical image. After several convolutions and pooling, the deep semantic information was not rich enough, and more details were lost. Moreover, the ResNet50 network had more layers, which slowed the detection speed. Therefore, we adopted a relatively shallow feature extraction network, that is, VGG16. We also changed the activation mode of each layer after convolution. The mAP was raised to 92.27% after the PReLU activation function was used. Compared with the original Faster R-CNN algorithm, the improved method enhanced the target recognition accuracy rate by 5.89%, recall rate by 7.11%, and mAP by 10.86%. It also reduced the false alarm number by 23 and the average detection time by 0.3138 s.

In the existing research, there is no theoretical expression to reflect the direct relationship between Faster R-CNN's final detection threshold (i.e., confidence) and the false alarm rate. Therefore, quantitative analysis is very difficult and can be obtained by the Monte Carlo simulation experiment, despite a huge amount of computation. In other words, in our application, there is indeed a certain relationship between confidence and the power of the noise plus sea clutter component. In the case of a certain confidence, the higher the power of the noise plus sea clutter component is, the greater the possibility of false-alarm

rate will be. Besides, the target detection mechanism is different between the CFAR detector and the proposed detector. For the CFAR detector, the target is detected when its amplitude exceeds certain threshold, while for the proposed detector based on Faster R-CNN, the target is detected by its morphological characteristics. Therefore, when the noise plus sea clutter component forms a certain shape which is similar to the shape of the target, the proposed detector may mistakenly the corresponding noise plus sea clutter component as a false alarm because the morphological characteristics of the target are learned by the proposed detector, which leads to the increase in the false-alarm rate.

## 5. Conclusions

This study proposed a target detection method of the coastal defense radar based on deep learning. Compared with the two-dimensional CFAR algorithm, the proposed method showed good anti-clutter and anti-jamming ability, higher recognition accuracy and recall rate, and fewer false alarms. Moreover, the average detection time per image was faster than that of the CA-CFAR and GOCA-CFAR. When multiple images must be detected, the speed difference between them was more obvious, meeting practical application in military reconnaissance and monitoring.

Some improvements on the feature extraction network, activation function, and RPN were made on the basis of the Faster R-CNN; hence, mAP was improved by 10.86%. This study made a novel attempt to apply deep learning to the target detection of the coastal defense radar. The experimental results verified the feasibility and effectiveness of the method. It is believed that in the near future, deep learning will solve more complex sea-surface target detection problems.

**Author Contributions:** Conceptualization, H.Y., C.C. and G.J.; methodology, H.Y., C.C., G.J. and J.Z.; software, H.Y. and X.W.; validation, G.J. and D.Z.; formal analysis, H.Y., X.W. and J.Z.; resources, H.Y., J.Z. and D.Z.; writing, H.Y. and C.C.; writing—review and editing, G.J. and X.W.; supervision, D.Z.; project administration, H.Y. and J.Z.; funding acquisition, H.Y. and D.Z. All authors have read and agreed to the published version of the manuscript.

**Funding:** This research was funded by the National Aerospace Science Foundation of China, grant number 201920052002.

**Conflicts of Interest:** The authors declare no conflict of interest.

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
