# Peer review of "Implementation of a Modified Faster R-CNN for Target Detection Technology of Coastal Defense Radar"

_remotesensing, doi:10.3390/rs13091703_

Round 1
Reviewer 1 Report
The manuscript presents a target detection method for coastal radar systems based on a Faster R-CNN algorithm. Following suggestions are provided for authors’ consideration.
- The mechanism of using Faster R-CNN algorithm to implement a CFAR detection has not been discussed clearly. For example, given a false alarm rate, how does the proposed method adjust to reach the design objective?
- The experimental details need to be supplemented. For example, how the training images have been chosen and what are the exact numbers samples are used for the train and the test experiments. For the train samples, how did the objects are labeled and how reliable this labeling is?
- For the performances shown in Table 2 and Figure 14 and 15, it is questionable that whether the parameters of the CFAR have been correctly set.
Reviewer 2 Report
The work reported here aims to develop a new target detection for the costal defense radar based on a modified version of faster region based CNN.
The paper deals with an interesting subject. The proposed methodology is clearly described. The provided experiments seem to prove good performances of the proposed version of Faster R-CNN target detector as compare as other DL-based methods.
However, the idea of using K-means clustering to initialize anchors for CNN-based detectors is not new:
Zhong, J. Wang, J. Peng and L. Zhang, "Anchor Box Optimization for Object Detection," 2020 IEEE Winter Conference on Applications of Computer Vision (WACV), Snowmass, CO, USA, 2020, pp. 1275-1283, doi: 10.1109/WACV45572.2020.9093498.
The authors should highlight their own contribution to the field.
Reviewer 3 Report
This paper describes the processing of the radars signals. Used references are actual and relevant. Article is easy for the reading.
The processing for the CA-CFAR is not very good, I don't see anything about the Doppler filtration (AMTI filters for suppression of the clutter - rain cells, static targets like islands and other), video integration (suppresion of the false alerts - very small or very big targets) and tracking of the targets. It are impotrant things for the radar signals preocessing. From the paper I can see that just the adaptive filtration (correlation) were used before the thrasholding and thresholded data were represented as the targets. It is not very good approach for the signals processing.
Check the figure 2, the line with the label x has not any source.
Use better resolution for the text in the figures
Chapters 4 and 5 haven't any introduction, after the chapter title is going immediately next chapter.
Make revision of the sentence on the line 248-249, the sentence looks weird.
The parameters for the used CA-CFAR in chapter 5 are missing (the N parameter) and did you tried more parameters or just one?
And why did you used CA-CFAR? Did you checked the characteristics of the different CFAR methods (CAGO-FAR, CASO-FAR, ...) and choose the optimal for your case? CA-CFAR is known that the results near the coast are very poor. And if you have the 2D image, which orientation did you used (if you used 1D CA-CFAR)? Why you didn't use any 2D thresholding method (maybe you did it, but you didn't write the parameters, so we cannot know)? But for the neural network you used 2D frames, so without any research I cen guess that the results for 2D method will be much better than for the 1D method. Now it is looking like you did the comparison of the very modern approach with approach old more than 30 years.
The case of the distant waters looks interesting for me, I am surprised when I see the line in the whole range, do you know what caused this event?
The final metod looks good with the good results, but for the comparison should me used more current method.
Round 2
Reviewer 1 Report
As I mentioned in my previous comment, the calculation of the false alarm rate of the manuscript has not been clearly presented. In the authors’ response, it is claimed that ‘the Faster R-CNN algorithm can use the confidence to ensure our expected false alarm rate’. Apart from little message is given in the revised manuscript, the false alarm rate, I believe, should be controlled by the noise rather by the detection side. Therefore, deeply worry may arise regarding its technique correctness for applying Fast RCNN to small target detection, particularly with a constraint of CFAR.
Reviewer 3 Report
The article looks very nice, I wish you good luck in your next researches.
Just the small things:
figure 2(now 3) I probably explained it wrong, the arrow in the algorithm (value x) has two output directions and no one input direction.
Please, add the reference for the AIS system principle.
And the sub-chaptures 2.1 and 2.3 needs introduction before the third level using.
